# Pharmacy and neighborhood-level variation in cash price of diabetes medications in the United States

Haider J. Warraich[1]*, Hasan K. Siddiqi[2], Diane G. Li[3], Jeroen van Meijgaard[3], Muthiah Vaduganathan[1]

1 Division of Cardiovascular Medicine, Department of Medicine, Brigham and Women's Hospital, Boston, MA, United States of America, 2 Division of Cardiovascular Medicine, Department of Medicine, Vanderbilt University Medical Center, Nashville, TN, United States of America, 3 GoodRx Inc, Santa Monica, CA, United States of America

* hwarraich@mgb.org

## Abstract

### Background

Diabetes medications place significant financial burden on patients but less is known about factors affecting cost variation.

### Objective

To examine pharmacy and neighborhood factors associated with cost variation for diabetes drugs in the US.

### Research design, subjects and measures

We used all-payer US pharmacy data from 45,874 chain and independent pharmacies reflecting 7,073,909 deidentified claims. We divided diabetes drugs into insulins, non-insulin generic medications, and brand name medications. Generalized linear models, stratified by pharmacy type, identified pharmacy and neighborhood factors associated with higher or lower cash price-per-unit (PPU) for each set of drugs.

### Results

Cash PPU was highest for brand name therapies ($149.4±203.2), followed by insulins ($42.4±25.0), and generic therapies ($1.3±4.4). Pharmacy-level price variation was greater for non-insulin generic therapies than insulins or brand name therapies. Chain pharmacies had both lower prices and lesser variation compared with independent pharmacies.

### Conclusions

Cash prices for diabetes medications in the US can vary considerably and that the greatest degree of price variation occurs in non-insulin generic therapies.

**Data Availability Statement:** GoodRx uses a representative sample of de-identified pharmacy claims across the United States, representing a wide range of channels and providers (not GoodRx

claims). These data are obtained through contractual arrangements with GoodRx's business partners (third party data). Data includes the drug and quantity that was dispensed, the location and date of the transaction, and the usual and customary cost (U&C) of the product that was dispensed. The U&C cost is what would be paid by the patient in the absence of a contracted rate, or insurance Price. These data do not include any personal identifiers, and do not include information on what was paid for the medication. These data are used to analyze trends in utilization and variation across geographic locales for specific drugs, classes of drugs, and medical conditions for which different drugs are prescribed. Use and disclosure of these data are restricted due to the specific terms and conditions in the contracts. Two authors of the paper, Diane G Li and Jeroen van Meijgaard, are employees of GoodRx and therefore have special privileges in accessing the data that other researchers would not have. Data access requests can be sent to legal@goodrx.com. In the subject line, requesters can reference the Research Team at GoodRx and the article title in question.

**Funding:** The author(s) received no specific funding for this work.

**Competing interests:** HW is a member of the GoodRx medical advisory board. DL and JvM are employees of GoodRx. MV has received research grant support or served on advisory boards for American Regent, Amgen, AstraZeneca, Bayer AG, Baxter Healthcare, Boehringer Ingelheim, Cytokinetics, Lexicon Pharmaceuticals, Novartis, Pharmacosmos, Relypsa, Roche Diagnostics, and Sanofi, speaker engagements with Novartis and Roche Diagnostics, and participates on clinical trial committees for studies sponsored by Galmed, Novartis, Occlutech, and Impulse Dynamics. This does not alter our adherence to PLOS ONE policies on sharing data and materials.

## Introduction

High medication costs place a significant burden on the health system and represent an important barrier to broad and equitable prescription drug access. Particular attention has been paid to the cost of medications used in diabetes care [1, 2], with a particular focus on the price of insulins.

Little is known about variation in diabetic drug costs based on pharmacy and neighborhood-level characteristics. This information would be important since both socioeconomic [3] and neighborhood level factors [4] are closely interlinked with population prevalence and disease outcomes in diabetes. Furthermore, significant variation in drug prices has been noted in prior work based on type of pharmacy (independent vs. chain) and neighborhood factors such as income and proportion of minority residents [5–7]. Black Americans experience greater financial burden from medications but are less likely to be aware of low-cost pharmacy options [8]. It would be important to assess the degree of variation in prices for diabetic medications, as well as to see what pharmacy or neighborhood level factors could be associated with higher or lower prices. Knowledge of these factors could 1) support patients seeking to lower drug costs, 2) provide policy makers with drivers of price variation to allow for the development of measures that could minimize variation, 3) ensure that vulnerable populations are not facing disproportionate price markups, and 4) could inform international audiences of potential factors affecting price variation that they could study in their local settings.

Therefore, we performed an analysis of a national pharmacy claims database linked with census data to examine pharmacy and neighborhood factors associated with cost variation for diabetes drugs in the US.

## Methods

We used all-payer US pharmacy data of chain and independent pharmacies across all 50 states [9]. These data reflect transactions that were submitted to health insurers and pharmacy benefit managers, with pharmacies reporting the usual and customary (U&C) price for the medication with each transaction. The U&C price is the lowest price a customer might have to pay in the absence of insurance and is also referred to as "cash price" [10]. This price is inclusive of all discounts. Pharmacies are required to report the U&C price which is also what they charge Medicare. Although most customers have prescription medication coverage and do not pay the U&C price, the U&C price is the most visible price at the retail level and impacts those lacking insurance. Pharmacies can respond to their customer needs by adjusting this price. Other prices such as the list price and the average wholesale price are constant across the country, while negotiated prices between PBMs and pharmacies are not publicly available and do not necessarily reflect what the customer ends up paying. The data were deidentified and therefore institutional review was not sought per HHS regulation 45 CFR 46.101(c).

Transactions between January 1st, 2019 to December 31st, 2019 were analyzed. We divided diabetes drugs into insulins (aspart, degludec, detemir, human, lispro, regular, glargine, glulisine), non-insulin generic therapies (metformin, sulfonylureas, thiazolidinediones), and brand name non-insulin therapies (glucagon-like peptide-1 receptor agonists [GLP-1RA], sodium glucose cotransporter-2 inhibitors [SGLT2i], dipeptidyl peptidase-4 inhibitors). Insulins included both generic insulins such as insulin aspart 70/30 and brand name insulins, which were the vast majority. Non-insulin generics did not include brand name formulations such as Glucophage. A total of 136 drug formulations were included in the analysis.

To allow for standardized comparisons across medications and dosages, for each drug claim we derived the cash price-per-unit (PPU) by dividing the reported usual and customary amount by the quantity filled. To aggregate the data from claim to pharmacy-level, we

averaged the cash PPU of a given drug across all claims in a particular pharmacy, weighted by the quantity filled on that claim. To exclude any extreme outliers, we grouped the data by drug, and determined a cutoff of 10x the median consistent with a prior published analysis from these data [6]. Because our model results examine percent changes, the interpretation of the results won't change even if we used cost per day as a measure. Our model also includes fixed effects at the drug-level which controls for the absolute level of price across drugs. By using drug-level fixed effects, we can use the model coefficients to estimate the cash PPU of drugs, and then multiply that by different doses and frequencies to estimate cost per day.

We then identified pharmacy and neighborhood factors associated with drug price variation. A generalized linear model with gamma distribution and log link function was constructed for each drug group and the models were stratified by pharmacy type (independent or chain). The cash price PPU data is a long right-skewed distribution. Log-link represented a natural choice for our cash price data, where residuals increase with x, and we expected errors to be proportional with the conditional mean. Our approach was derived using the methods outlined by Manning and Mullahy in their paper on estimating log models [11]. From the log link, we performed the modified park test, a modification of the heteroscedasticity test that is modified to help determine the family of distributions to use for a log link function. The test indicated that the Gamma family should be used.

Pharmacy-level cash PPU for a drug were regressed on ZIP code tabulation area (ZCTA) 3 level neighborhood variables and drug (fixed effects). Neighborhood variables sourced from the 2019 American Community Survey included mean household income (normalized), insurance coverage (Medicare, Medicaid, other insurance, no insurance coverage), race/ethnicity (Hispanic/Latino), and mean-to-median household income ratio. The proportion of housing units in urban areas was obtained from the 2010 US Decennial Census. To test for pharmacy type interaction, a pooled model was run, interacting the pharmacy type variable with all neighborhood variables, but without interaction with drug (fixed effects). We cross-walked ZIP codes within the 50 states from 5 (ZCTA5) using the 2019 Uniform Data System Mapper ZIP code to ZCTA Crosswalk and used the first 3 digits of the ZCTA5 to group data at the ZCTA3 level. To interpret the model results for neighborhood-level variables across ZCTA3s, we multiplied model coefficients for each neighborhood variable by its inter-quartile range and then exponentiating it. The resulting number is the % change of the cash PPU going from the 1st to the 3rd quartile for a given neighborhood variable. A cash price per PPU change greater than 5% from 1st to 3rd quartile was considered noteworthy.

PPU is used to standardized the analysis, and sufficient for calculating % variation across ZCTA3. PPUs are insufficient for estimating the cost of a prescription due to difference in form, e.g. a unit of insulin can be used over multiple days, while a tablet, also a unity, may be taken multiple times a day. Monthly estimated costs for a typical prescription for one common drug from each drug group were calculated for interpretability of magnitude of costs and variation across.

To visualize degree of pharmacy-level variation, we generated scatterplots of drug pricing across ZCTA3s, excluding ZCTA3s with 20,000 or fewer total claims in 2019. We created a ZCTA3-drug group level price index to determine if a ZCTA3 had prices higher or lower than the national average for a particular group of drugs. For each ZCTA3, we calculated the price index by taking the average cash PPU of each drug and dividing it by the average PPU nationally. To aggregate the price index to the ZCTA3-drug group level, we took the average of each drug's price index in a given grouping, weighted by the proportion of the total revenue of that drug nationally to the total revenue of all drugs in our cohort nationally. Python (Version 3.8) and STATA (Version 16.1) were used to perform the statistical analysis. Lastly, this was an investigator-initiated research proposal that was analyzed by the health economics team at Goodrx.

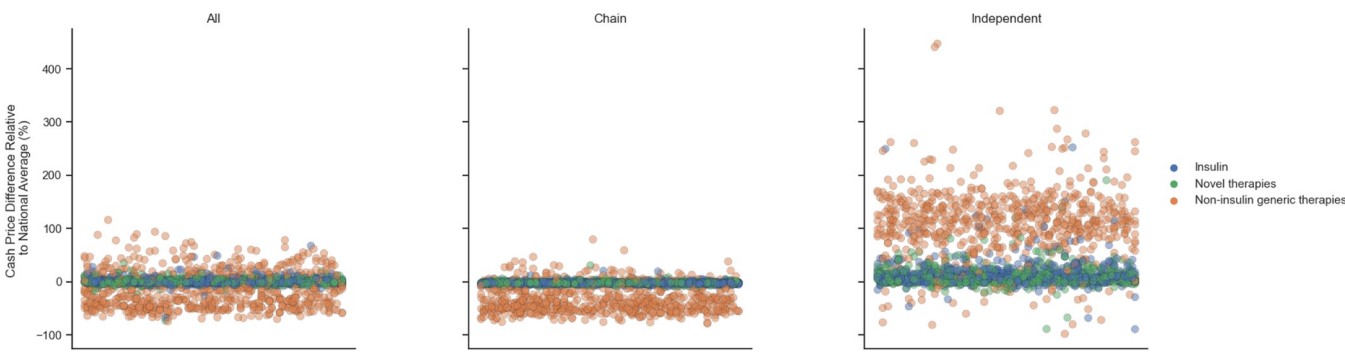

**Fig 1. Price variation in diabetes medications by pharmacy type.** Each circle represents the cash prices in a particular ZCTA3 relative to the national average for that drug group: blue for insulin, orange for generic therapies, and green for brand name therapies. Y axis represents the cash price index where 0 is the average cash price nationally. X axis arranges individual pharmacies serially.

## Results

Our sample included 45,874 pharmacies (11,927 independent, 33,947 chain), across 12,883 ZIP codes and 882 ZCTA3s, reflecting 7,073,909 deidentified claims. The average mean household income across ZIP codes was $80,592.1±24,201.5. Average proportion of the population covered by commercial insurance was 53%, followed by Medicaid (18%), Medicare (13%), other insurance (8%) and no insurance (8%). Average proportion of the population that was non-Hispanic White was 70%, and 66% of housing units were in urban areas. The mean cash PPU was highest for brand name therapies ($149.4±203.2), followed by insulins ($42.4±25.0), and generic therapies ($1.3±4.4). Pharmacy-level price variation was higher for generics than insulins or brand name therapies (**Fig 1**).

Several neighborhood variables were significantly associated with change in cash prices per PPU when comparing the 1st quartile and the 3rd quartile for that covariate within each pharmacy stratum (**Table 1**). Insulin cash prices were lower in independent pharmacies with greater proportion of White residents (-6.7%) and urban housing units (-7.6%) (both p<0.001). Brand name prices were lower in areas with more urban housing units (-7.0%, p<0.001). Chain pharmacies had both lower mean prices and lesser variation compared to independent pharmacies.

The greatest degree of price variation between the 1st and 3rd quartile of the neighborhood covariates analyzed was noted for generic therapies. Chain pharmacies were likely to have higher prices in neighborhoods with greater proportions of high-income residents (+7.9%), urban housing units (+8.4%), and Medicaid beneficiaries (+17.3%), and had lower prices in neighborhoods with greater proportion of people without insurance (-11.2%) (all p<0.001). Independent pharmacies had higher prices in neighborhoods with greater proportion of Medicare beneficiaries (+7.5%) and lower prices in neighborhoods with more White residents (-8.7%) (all p<0.001). Similar trends were noted when assessing 30-day supply for most common prescriptions: at independent vs chain pharmacies, glargine insulin cost $585.6 (532.9–707.8) vs $494.3 (482.3–513.9), metformin cost $44.2 (34.5–60.8) vs $16.8 (10.6–28.5), and empagliflozin cost $659.2 (574.3–781.1) vs $592.8 (571.4–608.2) respectively.

## Discussion

Prices for diabetes medications can vary considerably and the greatest degree of price variation occurs in non-insulin generic therapies. Independent pharmacies have higher prices on average and greater price variation compared with chain pharmacies.

**Table 1. Pharmacy and neighborhood factors associated with cost variation in diabetes medications.**

| Coefficients | Insulins | | Non-Insulin Generics | | Brand Name Therapies | |
|---|---|---|---|---|---|---|
| | Chain (n = 136,743) | Independent (n = 40,043) | Chain (n = 239,242) | Independent (n = 54,099) | Chain (n = 103,597) | Independent (n = 30,747) |
| **Household Income** | -0.00 | 0.01 | 0.08*** | 0.03*** | -0.00 | 0.03*** |
| | [-0.00,0.00] | [-0.00,0.03] | [0.06,0.09] | [0.01,0.05] | [-0.00,0.00] | [0.01,0.04] |
| | -0.1% | 1.1% | 7.9% | 3.0% | -0.1% | 2.7% |
| **Mean to Median Income Ratio** | -0.01 | 0.03 | -0.17*** | 0.11* | -0.02*** | -0.03 |
| | [-0.02,0.00] | [-0.08,0.14] | [-0.26,-0.08] | [0.01,0.21] | [-0.03,-0.01] | [-0.12,0.06] |
| | -0.1% | 0.3% | -1.7% | 1.1% | -0.2% | -0.3% |
| **Non-Hispanic White** | 0.03*** | -0.22*** | -0.03 | -0.29*** | 0.02*** | -0.15*** |
| | [0.02,0.04] | [-0.31,-0.13] | [-0.11,0.04] | [-0.38,-0.21] | [0.01,0.03] | [-0.22,-0.07] |
| | 0.8% | -6.7% | -1.1% | -8.7% | 0.7% | -4.5% |
| **Medicare** | -0.05* | 0.42 | 1.11*** | 1.88*** | -0.01 | 0.52** |
| | [-0.08,-0.01] | [-0.01,0.86] | [0.81,1.42] | [1.35,2.40] | [-0.04,0.02] | [0.13,0.91] |
| | -0.2% | 1.7% | 4.4% | 7.5% | -0.0% | 2.0% |
| **Medicaid** | 0.09*** | -0.09 | 1.86*** | 0.45*** | 0.09*** | 0.10 |
| | [0.06,0.11] | [-0.32,0.14] | [1.65,2.06] | [0.20,0.70] | [0.07,0.12] | [-0.10,0.29] |
| | 0.8% | -0.8% | 17.3% | 4.0% | 0.8% | 0.8% |
| **Other Insurance** | -0.09*** | 0.60* | -0.67*** | 1.44*** | -0.09*** | 1.00*** |
| | [-0.13,-0.06] | [0.14,1.05] | [-0.97,-0.38] | [0.99,1.89] | [-0.12,-0.05] | [0.55,1.46] |
| | -0.3 | 1.6% | -1.8% | 4.0% | -0.2% | 2.8% |
| **No Insurance** | -0.11*** | 0.19 | -2.31*** | 0.39 | -0.17*** | 0.36* |
| | [-0.15,-0.08] | [-0.16,0.53] | [-2.60,-2.02] | [-0.02,0.80] | [-0.20,-0.14] | [0.01,0.70] |
| | -0.6% | 1.0% | -11.2% | 2.0% | -0.9% | 1.9% |
| **Urban Housing Units** | -0.01 | -0.16*** | 0.16*** | -0.02 | -0.00 | -0.15*** |
| | [-0.01,0.00] | [-0.22,-0.10] | [0.11,0.22] | [-0.09,0.05] | [-0.01,0.00] | [-0.20,-0.09] |
| | -0.3% | -7.6% | 8.4% | -0.9% | -0.2% | -7.0% |

Positive values represent cash prices higher than average and negative values present cash prices lower than average. Percentages represent difference in cash price per unit between 1st and 3rd quartile of neighborhood factors.

N represents pharmacy-drug groups. 95% confidence interval in brackets.

*p<0.05

**p<0.01

***p<0.001.

Several neighborhood level factors appeared to influence pricing. The most significant association noted was that chain pharmacies had much lower prices in neighborhoods where lack of insurance was more common. This is important because even though high cash prices in general lead to higher drug costs for the entire system, cash prices most directly affect patients without insurance. Chain pharmacies were likely to have higher prices in neighborhoods where affordability was less likely to be an issue such as in neighborhoods with greater household income. On the other hand, independent pharmacies had lower prices for all categories, including insulins and novel therapies, in areas with a higher proportion of White residents. The exact mechanisms underlying this neighborhood and pharmacy-based price variation are uncertain though it would suggest that chain pharmacies are more sensitive to neighborhood-level factors. Another possible explanation may be that larger, chain pharmacies have the resources to reach specific customer groups through marketing and pricing strategy such as discount programs. Chain pharmacies may also have negotiating power to purchase drugs at a lower price and offer lower cash prices to uninsured patients.

While our study analyzes cash prices, which most directly affect patients lacking insurance, cash prices also affect patients with health insurance. Many patients with high-deductible health insurance plans have to pay the list prices until they hit their deductible and higher list prices are often associated with higher out of pocket costs for many privately insured patients [12].

Several limitations of this analysis should be highlighted. We dichotomized pharmacy types into independent and chain [5], though future work could further characterize pharmacy chains based on size (indicating purchasing power), type (grocery-based vs. big box), and level of decision making (national vs. regional/local).

These findings reveal that simply the pharmacy that patients use can introduce significant variation in price. While traditionally the focus of drug costs for diabetics has been on insulins or brand name therapies, we find that the price of generic therapies can vary considerably. This is important because even though generic therapies are generally cheaper than other therapies, this significant variation suggests that they are the most sensitive to neighborhood-level variation and that even patients using these therapies could face high financial burden based on where they acquire their generic therapies from. Patients facing financial burden from healthcare costs may consider searching for an accessible pharmacy that might offer them the best rates. Our findings suggest this might be particularly true for chain pharmacies that serve regions with a large proportion of patients lacking health insurance, who are more likely to pay cash prices. In addition, policy makers may increase scrutiny of pharmacies that consistently charge higher prices and study the effect of such markups on healthcare costs.

## Acknowledgments

This article was written prior to Dr Warraich joining the US Food and Drug Administration. None of the views expressed herein represent those of the FDA or the US Government.

## Author Contributions

**Conceptualization:** Haider J. Warraich, Muthiah Vaduganathan.

**Data curation:** Diane G. Li.

**Formal analysis:** Diane G. Li, Jeroen van Meijgaard.

**Investigation:** Haider J. Warraich, Hasan K. Siddiqi, Diane G. Li, Jeroen van Meijgaard, Muthiah Vaduganathan.

**Methodology:** Haider J. Warraich, Diane G. Li, Jeroen van Meijgaard, Muthiah Vaduganathan.

**Project administration:** Haider J. Warraich, Jeroen van Meijgaard.

**Resources:** Haider J. Warraich, Diane G. Li, Jeroen van Meijgaard.

**Software:** Diane G. Li, Jeroen van Meijgaard.

**Supervision:** Haider J. Warraich, Jeroen van Meijgaard, Muthiah Vaduganathan.

**Validation:** Diane G. Li, Jeroen van Meijgaard.

**Visualization:** Diane G. Li.

**Writing – original draft:** Haider J. Warraich, Hasan K. Siddiqi, Muthiah Vaduganathan.

**Writing – review & editing:** Haider J. Warraich, Hasan K. Siddiqi, Diane G. Li, Jeroen van Meijgaard, Muthiah Vaduganathan.

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
