## [Decision Letter · Decision Letter 0]

5 Jun 2023

PONE-D-23-00503Pharmacy and Neighborhood-Level Variation in Price of Diabetes Medications in the United StatesPLOS ONE

Dear Dr. Warraich,

Thank you for submitting your manuscript to PLOS ONE. After careful consideration, we feel that it has merit but does not fully meet PLOS ONE’s publication criteria as it currently stands. Therefore, we invite you to submit a revised version of the manuscript that addresses the points raised during the review process.

We look forward to receiving your revised manuscript.

Kind regards,

Mihajlo Jakovljevic, MD, PhD, MAE

Academic Editor

PLOS ONE

Journal Requirements:

"..MV has received research grant support or served on advisory boards for American Regent, Amgen, AstraZeneca, Bayer AG, Baxter Healthcare, Boehringer Ingelheim, Cytokinetics, Lexicon Pharmaceuticals, Novartis, Pharmacosmos, Relypsa, Roche Diagnostics, and Sanofi, speaker engagements with Novartis and Roche Diagnostics, and participates on clinical trial committees for studies sponsored by Galmed, Novartis, Occlutech, and Impulse Dynamics.."

3. Thank you for stating the following in the Competing Interests:

"HW is a member of the GoodRx medical advisory board. DL and JvM are employees of GoodRx. MV has received research grant support or served on advisory boards for American Regent, Amgen, AstraZeneca, Bayer AG, Baxter Healthcare, Boehringer Ingelheim, Cytokinetics, Lexicon Pharmaceuticals, Novartis, Pharmacosmos, Relypsa, Roche Diagnostics, and Sanofi, speaker engagements with Novartis and Roche Diagnostics, and participates on clinical trial committees for studies sponsored by Galmed, Novartis, Occlutech, and Impulse Dynamics."

We note that one or more of the authors have an affiliation to the commercial funders of this research study : GoodRx, American Regent, Amgen, AstraZeneca, Bayer AG, Baxter Healthcare, Boehringer Ingelheim, Cytokinetics, Lexicon Pharmaceuticals, Novartis, Pharmacosmos, Relypsa, Roche Diagnostics, Sanofi, Roche Diagnostics, Galmed, Occlutech, and Impulse Dynamics.

(2) Please also provide an updated Competing Interests Statement declaring this commercial affiliation along with any other relevant declarations relating to employment, consultancy, patents, products in development, or marketed products, etc.  

Within your Competing Interests Statement, please confirm that this commercial affiliation does not alter your adherence to all PLOS ONE policies on sharing data and materials by including the following statement: ""This does not alter our adherence to  PLOS ONE policies on sharing data and materials.” (as detailed online in our guide for authors http://journals.plos.org/plosone/s/competing-interests). If this adherence statement is not accurate and  there are restrictions on sharing of data and/or materials, please state these. Please note that we cannot proceed with consideration of your article until this information has been declared.

Reviewers' comments:

Reviewer's Responses to Questions

**Comments to the Author**

1. Is the manuscript technically sound, and do the data support the conclusions?

Reviewer #1: Partly

Reviewer #2: Partly

2. Has the statistical analysis been performed appropriately and rigorously? 

Reviewer #1: No

Reviewer #2: Yes

3. Have the authors made all data underlying the findings in their manuscript fully available?

Reviewer #1: No

Reviewer #2: No

4. Is the manuscript presented in an intelligible fashion and written in standard English?

Reviewer #1: Yes

Reviewer #2: Yes

5. Review Comments to the Author

Reviewer #1: This study aimed to examine pharmacy and neighborhood factors associated with cost variation for diabetes drugs in the US using all-payer US pharmacy data from 45,874 chain and independent pharmacies reflecting 7,073,909 deidentified claims. The authors reported that cash prices for diabetes medications in the US can vary considerably and that the greatest degree of price variation occurs in non-insulin generic therapies. Overall, it’s topically interesting and timely as the authors properly stated that “high medication costs place a significant burden on the health system and represent an important barrier to broad and equitable prescription drug access.” The reviewer has four main comments regarding conceptualization, methodology, interpretation of the results, and limitations.

1. Conceptualization: the authors stated that “The U&C price is the price charged to a customer in the absence of insurance or negotiated discount. Although most customers have prescription medication coverage and do not pay the U&C price, the U&C price is the most visible price at the retail level and impacts those lacking insurance. “(Methods, Page 4) Conceptually, the U&C price is not the price a consumer pays for the medication if the consumer has health insurance, and most US consumers have certain insurance coverage such as Medicare, Medicaid, and private insurance. Hence, the “price” the authors referred to was not the actual price most consumers were paying, and its implication to practice and policy is unclear. A well-thought-out introduction of the U&C price in the light of transactions among the wholesaler, pharmacy chains, consumers with or without insurance is needed to help the readers to better understand the relevance and significance of the study.

2. Methodology: the authors stated that “A generalized linear model with gamma distribution and log link function was constructed for each drug group and the models were stratified by pharmacy type (independent or chain) (Methods, Page 5). Were family (gamma) and link (log) tests performed to determine the appropriateness of the gamma distribution and log link function? How did the distribution of cash price-per-unit (PPU) look like? Some tables/figures would be informative to show the distribution of the outcome measure the authors had in the analysis by drugs.

3. Interpretation of the results: the authors stated that “We then identified pharmacy and neighborhood factors associated with drug price variation” using the generalized linear model (Methods, Page 5). The authors also stated that “Neighborhood variables sourced from the 2019 American Community Survey included mean household income (normalized), insurance coverage (Medicare, Medicaid, other insurance, no insurance coverage), race/ethnicity (Hispanic/Latino), and mean-to-median household income ratio.” In Results, the analysis was then interpreted as “Average proportion covered by commercial insurance was 53%, followed by Medicaid (18%), Medicare (13%), other insurance (8%) and no insurance (8%).” (Results, Page 6). Was this insurance coverage derived from the ACS or the pharmacy data? Please clarify. Please also clarify the unit of analysis, for example, how the neighborhood was defined, how many neighborhoods were covered in this data, and the reference groups in the results, for example, when the authors stated that “Chain pharmacies were likely to have higher prices in neighborhoods with greater proportions of high-income residents (+7.9%), urban housing units (+8.4%), and Medicaid beneficiaries (+17.3%), and had lower prices in neighborhoods with greater proportion of people without insurance (-11.2%),” (Results, Page 7), please clarify the reference groups for each of these comparisons, and where those coefficients came from, their associated p-values, for example, was this +7.9% associated with each unit increase in the household income by ACS, if so, what is the unit increase of household income?

4. Limitation: While the authors stated that “Pharmacy-level price variation was greater for non-insulin generic therapies than insulins or brand name therapies. Chain pharmacies had both lower prices and lesser variation compared with independent pharmacies”, (Results, Abstract), their implication to practice and policy was unclear given the generic drugs had much lower acquisition costs and cash prices, hence, the cost burden to those customers who do not have health insurance is much smaller. In addition, since the U&C price is not the actual price most consumers are paying, it would be more appropriate to span out the “usual and customary prices” or “cash price” in the title to clarify.

Reviewer #2: I have also uploaded my comments: 1. Authors gave a good rationale for the study- differences in prices could affect those without insurance, other cash paying patients and possibly those with deductibles. I have some comments for clarification and/or consideration.

2. Can you give a reference for the definition of U&C, I saw a reference by Joey Mattingly that the law 42C.F.R &447.512(b) requires pharmacies to report the U&C when submitting pharmacy claims for both Rx and OTC but not cash paying sales?

3. The authors divided the diabetic medications into 3 groups: insulins, non-insulin generics, and brand name non-insulins. What about brand name drugs that are not GLP1s, SGLTs and DPP4s, such as Glucophage? Or brand name insulins? Were they eliminated from the study?

4. I think cash PPU is a poor measure to compare these drugs since collectively, they are given at different doses and frequencies (some are once a week, twice a day etc) why don’t you ‘cost per day’? Or by class, such as generic biguanides, brand name biguanides, GLP1s etc? The variability will be significantly different.

5. Can you define chain pharmacies? Do you mean stand-alone pharmacies like CVS or does it include supermarket chains like Walmart? These 2 chain-types have different pricing strategies and should be treated differently.

6. Can you add the rationale for choosing cutoff for extreme outliers at 10X the median?

7. Do you have a reference for how you treated variability across neighborhoods on page 5?

8. Typo: Tablet instead of table (page 5 last line)

9. Please use generic names instead of brand names like Lantus and Jardiance.

10. Because of how broadly you classified the drugs into 3 groups- it weakens the relevance of the findings. It will be better to treat the drugs by class and that will be meaningful for prescribers.

6. PLOS authors have the option to publish the peer review history of their article (what does this mean?). If published, this will include your full peer review and any attached files.

Reviewer #1: No

Reviewer #2: No

---

## [Author Response · Author response to Decision Letter 0]

20 Jul 2023

Please see attached response to reviewers for complete response

---

## [Decision Letter · Decision Letter 1]

27 Oct 2023

Pharmacy and Neighborhood-Level Variation in Cash Price of Diabetes Medications in the United States

PONE-D-23-00503R1

Dear Dr. Warraich,

We’re pleased to inform you that your manuscript has been judged scientifically suitable for publication and will be formally accepted for publication once it meets all outstanding technical requirements.

Kind regards,

Mihajlo Jakovljevic, MD, PhD, MAE

Academic Editor

PLOS ONE

Additional Editor Comments (optional):

Reviewers' comments:

Reviewer's Responses to Questions

**Comments to the Author**

1. If the authors have adequately addressed your comments raised in a previous round of review and you feel that this manuscript is now acceptable for publication, you may indicate that here to bypass the “Comments to the Author” section, enter your conflict of interest statement in the “Confidential to Editor” section, and submit your "Accept" recommendation.

Reviewer #1: All comments have been addressed

2. Is the manuscript technically sound, and do the data support the conclusions?

Reviewer #1: Yes

3. Has the statistical analysis been performed appropriately and rigorously? 

Reviewer #1: Yes

4. Have the authors made all data underlying the findings in their manuscript fully available?

Reviewer #1: Yes

5. Is the manuscript presented in an intelligible fashion and written in standard English?

Reviewer #1: Yes

6. Review Comments to the Author

Reviewer #1: The authors have been responsive to reviewers' comments. The manuscript is stronger with greater clarity.

7. PLOS authors have the option to publish the peer review history of their article (what does this mean?). If published, this will include your full peer review and any attached files.

Reviewer #1: No

---

## [Editor Report · Acceptance letter]

27 Nov 2023

PONE-D-23-00503R1 

Pharmacy and Neighborhood-Level Variation in Cash Price of Diabetes Medications in the United States 

Dear Dr. Warraich:

I'm pleased to inform you that your manuscript has been deemed suitable for publication in PLOS ONE. Congratulations! Your manuscript is now with our production department. 

Kind regards, 

on behalf of

Professor Mihajlo Jakovljevic 

Academic Editor

PLOS ONE